# *Selflife*: A Life Skills Development Tool to Prevent Sexual Violence among Healthcare Students

**DOI:** 10.3390/ijerph20065198

**Published:** 2023-03-15

**Authors:** Sylvie Pires, Hélène Denizot, Abdel Halim Boudoukha, Julie Mennuti, Cécile Miele, Catherine Potard, Gaëlle Riquoir, Pierre-Michel Llorca, Valentin Flaudias, Laurent Gerbaud

**Affiliations:** 1CRIAVS ARA, Délégation de Clermont-Ferrand, Pôle Santé Publique, CHU Gabriel Montpied, 63000 Clermont-Ferrand, France; 2Pôle de Psychiatrie B, CHU Gabriel Montpied, 63000 Clermont-Ferrand, France; 3Nantes Université, Univ Angers, Laboratoire de psychologie des Pays de la Loire, LPPL, UR 4638, 44000 Nantes, France; 4Service de Santé Publique, CHU de Clermont-Ferrand, Université Clermont Auvergne, CNRS-UMR 6602, Institut Pascal, Axe TGI, 63011 Clermont-Ferrand, France; 5Pôle de Psychiatrie B, CHU de Clermont-Ferrand, Université Clermont Auvergne, CNRS-UMR 6602, Institut Pascal, Axe TGI, 63011 Clermont-Ferrand, France

**Keywords:** sexual violence, prevention, trauma, students, life skills

## Abstract

Background: in France, 14.5% of women and 3.9% of men aged 20–69 years have experienced sexual violence. Of these, 40% will go on to develop posttraumatic stress disorder. Sexual violence is therefore a major public health issue. In the present study, we tested a life skills development tool (*Selflife*) designed to prevent sexual violence in a population of healthcare students. Methods: a total of 225 French healthcare students were randomly divided into a control group using case studies (*n* = 114) and a group using *Selflife* (*n* = 111) to discuss the topic of sexual violence. After the session, they completed self-report questionnaires collecting sociodemographic data and probing their feelings about their participation, their life skills, and their verdict on the intervention. Results: compared with controls, participants in the *Selflife* group reported gaining more knowledge about sexual violence, a greater sense of improving their life skills, and greater satisfaction with the intervention. Conclusions: these results suggest that, in addition to providing information about sexual violence, *Selflife* helped students develop their life skills, thereby empowering them to act when confronted with sexual violence. Its impact on prevalence and on the psychological and psychiatric consequences remains to be assessed.

## 1. Introduction

In France, 14.5% of women and 3.9% of men aged 20–69 years have experienced sexual violence [1]. In this study, sexual violence is defined as any act related to rape, attempted rape, sexual assault, or coercion related to sexual acts (e.g., being forced to watch sex films). These events could have occurred in the context of school, work, public spaces (broadly defined), in the context of a couple or ex-spouse relationship (including boyfriends or non-cohabiting relationships), and in the context of family and close relationships. Excluding harassment and exhibitionism. Sexual violence is a traumatic event, and meets Criterion A (exposure to actual or threatened death, serious injury, or sexual violence) of posttraumatic stress disorder (PTSD) in the Diagnostic and Statistical Manual of Mental Disorders, Fifth Edition [2]. Among the victims of sexual violence, 40% go on to develop PTSD [3]. Given its prevalence, sexual violence is therefore a major public health problem [4,5] which persists despite enhancement of research in this field (three times as many articles in 2022 than 2010 about “sexual violence” on Google Scholar) or political responses (such as the European program DAPHNE, which aims to reduce gender violence and has a budget of EUR 80 million for the period 2021–2027 [6,7]). 

### 1.1. Sexual Violence among Students 

Various published reports and surveys show that sexual violence frequently begins in adolescence and escalates in emerging adulthood up to the age of 24–25 years [8]. As a result, a significant proportion of sexual violence occurs within the student population with testimonies and figures that are quite disturbing, especially since the emergence of movements such as #metoo encouraging women to speak out. For example, in the study of Williams et al. [8], where they explored unwanted sexual activities due to pressure (by friendship or romantic relationship) or another person threatened to use or did use physical force, or when they did not want to because they were drunk or on drugs, they observed that 12.45% of students had experienced a case of sexual violence in the last 12 months. Thus, in France, 1 in 10 women claim to have been sexually assaulted during their time at university, 1 in 20 students claim to have been raped, and 34% of male and female undergraduates claim to have been victims or witnesses of sexual violence [9].

In nine cases out of ten, the victims know their aggressor, who can be a boyfriend, romantic partner, or another student [10]. In the majority of cases, the acts would have taken place in a festive context or within the student residence [11]. Adherence to rape myths encourages sexual violence, while minimizing the responsibility and guilt of the perpetrators [12,13,14,15]. Some men may interpret a woman’s friendly interest as sexual interest, her lack of consent as feigned resistance, and her stunned response to an assault as a form of tacit acceptance. Studies also show a strong link between being sexually abused as a child or teenager and becoming a perpetrator in turn [16,17]. There is probably no typical portrait of a perpetrator of sexual violence. Young adult students are at a particularly high risk of violence. The combination of all three types of factors predicts the severity of aggression.

In medical schools, which are highly competitive environments, the prevalence is supposedly even higher [18]. More specifically, 8.6% of medical students report having experienced sexual violence. Given the uniquely stressful nature of medical school, the health of medical students has been explored in many countries [19], including France, where a report [4] highlighted the very severe consequences of sexual violence in terms of health and future lives, including addictive, suicidal behaviors and posttraumatic stress symptoms.

The student population in the 18–25-year age range is known to be particularly vulnerable to psychopathological disorders such as anxiety, depressive symptoms, eating disorders, and alcohol abuse. It has been shown that 75% of mental health disorders develop before the age of 25 years [20], not least because poverty is especially acute among young people [21].

Furthermore, the general public lack knowledge and adequate information about sexual violence, which often remains misunderstood or even trivialized in certain social representations. Social representations refer to the beliefs and judgments we construct to represent ourselves in the social world [22]. They “circulate in discourse and are carried by words, conveyed in messages and media images, and crystallized in behaviors and material or spatial arrangements” [23]. As they are registered in our cognitive schemas [24], social representations can influence the way we select, process, and memorize the information around us. Together with our knowledge and beliefs, social representations form our cognitions, which interact with our behaviors and emotions. They may be one reason why sexual violence is not always identified as such and is therefore under-reported, as suggested in a recent French report [25].

### 1.2. Acting on Representations to Prevent Sexual Violence

Social representations related to sexual practices have a major impact on behaviors and whether or not an act of sexual violence is actually perpetrated. In other words, if we are incapable of imagining what sexual violence is, we cannot spot risky situations or identify acts that fall under this heading. This at least is what the various surveys on this subject have shown. In 2019, for instance, one in two French people considered that forcing someone to perform oral sex does not constitute rape [25]. There is still a great deal of ignorance surrounding sexual violence and what the law says about it. Adherence to false representations of rape (the concept of rape myth is often used to describe these false representations/beliefs) could explain the persistence of violence despite the various information campaigns that have been rolled out to date. Moor [26] reported that women who adhere to a rape myth feel responsible for the assault, are less likely to report it, and are less likely to seek help, making them even more vulnerable. In addition, Voller et al. [27] pointed out that acceptance of the rape myth and the devaluation of emotions are directly associated with low self-efficacy, which in turn is related to more severe symptoms. Finally, Harned [28] observed that female victims who did not label what they had experienced as rape or sexual assault, despite feeling victimized, viewed the event as less severe.

### 1.3. Prevention Programs

It is therefore extremely important to focus on providing more information and implementing more prevention programs among the student population. The twofold objective of this type of initiative should be to prevent risky behaviors leading to sexual violence, and to equip participants with efficient strategies for coping with sexual aggression. Moreover, sexual violence prevention efforts must focus on both the victims and the potential perpetrators [17]. To achieve this, work must be conducted on learning about the conditions under which sexual violence occurs, on social representations, and finally, on developing the skills to deal with an experience of sexual violence.

It does not seem appropriate to draw inspiration from prevention programs that focus solely on providing information about the risks involved, as these types of program have long been shown to have limitations [29,30]. Several program designers have recently tried to consider the problem differently, focusing on ways of reinforcing people’s strategies for coping with social pressure, failure, or stress [31].

It was with this in mind that the Auvergne Resource Center for Professionals Working with Perpetrators of Sexual Violence (CRIAVS) set about developing a tool (*Selflife*) to empower healthcare students who encounter sexual problems by strengthening their life skills (for a more specific description, see paragraph by Canale and Mennuti in Miele and Canale, [32]). This tool takes the form of a board game, with several cards and themes relating to the issues of sexual violence. This fun approach makes it possible to evoke both problems and representations, while developing life skills. In 1993, the World Health Organization identified several life skills as being beneficial for health, and it seemed important to embed them in *Selflife*. They were originally listed in five pairs: problem solving–decision making, creative thinking–critical thinking, effective communication–interpersonal skills, self-awareness–empathy, and emotion regulation–stress management.

These life skills help prevent mental health problems, substance use, and risky behaviors [33], and can be viewed as coping strategies for reducing the severity of traumatic experiences (especially the emotion regulation–stress management pair).

### 1.4. Objectives

Given the prevalence of sexual violence and its psychological and physical repercussions, it is important to implement programs to prevent and promote students’ sexual health in a safe environment. The purpose of the present study was to undertake a preliminary assessment of the *Selflife* tool in a population of healthcare students.

We hypothesized that the *Selflife* intervention would be associated with (a) a feeling of acquiring knowledge about the social representations of sexual violence at least equivalent to traditional teaching; (b) a better development of their life skills; and (c) a better satisfaction with their participation in this teaching. For this, we used a comparison postintervention between an intervention using *Selflife* and an intervention using the presentation of case studies, which is already an effective teaching modality in medical education [34].

## 2. Materials and Methods

### 2.1. Participants and Procedures

A total of 225 students (165 (73.7%) women; *M*_age_ = 21.4 years, *SD* = 3.2) enrolled on a compulsory module on the prevention of sexual violence at the University of Clermont Auvergne volunteered to take part in the present study. We used cluster randomization to assign them either to a group using case studies or to a group using the *Selflife* tool. In the case studies group, participants were asked to think about various situations potentially involving sexual violence. They had to say whether sexual violence was indeed involved and think about possible factors/resources in each one. The procedure for using the *“Selflife”* tool is described below in Section 2.3 (“Description of *Selflife* Tool”). At the end of this single-session intervention, each student completed a set of questions collecting their sociodemographic data and their feelings about their participation (i.e., knowledge gained, development of life skills, and satisfaction with the intervention). The whole process lasted 2 h. The research was carried out in accordance with The Code of Ethics of the World Medical Association (Declaration of Helsinki) for experiments involving humans. Authorizations were obtained by the institutions where the study was conducted. No specific number was assigned. The course leader (L.G.) has accepted this evaluation and data collection. All the participants provided a valid informed consent before the beginning of the study.

### 2.2. Measures

We created a self-report questionnaire to probe participants’ feelings about their knowledge acquisition. Participants rated each of the five items on a scale ranging from 0 (Not at all) to 10 (Strongly agree).

Participants were also asked about the 10 life skills. They had to indicate how well they felt they had developed each one during the intervention on a scale ranging from 0 to 10 (“For each of the life skills listed below in the table, circle the number corresponding to your feeling”).

Lastly, participants were asked to indicate their level of satisfaction with the intervention by rating six statements on a Likert-like scale ranging from 1 to 4 (Very satisfied, Satisfied, Moderately satisfied, Not at all satisfied, respectively).

### 2.3. Description of Selflife Tool

*Selflife* is a fun mediation tool with a preventive aim. It takes the form of a goose game. Its primary objective is to support elaboration and exchange, by encouraging participants to (a) think about risky situations, or those that can generate discrimination, violence, and suffering, as well as the factors that facilitate them, and (b) work on their social representations, which are sometimes far removed from reality. Another objective is to improve life skills. The game’s various themes are therefore based on the 10 life skills. For example, before working on a particular theme, participants are asked to “put forward solutions” (referring to the skill of problem solving).

The squares on the board feature statements on which the players are invited to give their opinion, short stories to complete, surprise cards to discuss, or a choice of cards, where participants select the scenario that most closely matches what they think or know, thereby placing them in the position of actors. These cards deal with color-coded topics (aniseed green: sexuality/making love; pomegranate red: violence/transgression; petroleum blue: partying/being in a group; mint: real/virtual exposure on social media; tangerine: self-image and self-esteem/norms).

### 2.4. Description of Case Studies Course

For the “case studies” teaching method, we chose to present examples of sexual violence in the student population based on facts reported in the national press. The objective was also to encourage exchanges within the group of students about the case presented, while mobilizing their psychosocial skills, by developing their critical thinking skills, for example, or by probing them about possible resources in the face of sexual violence.

### 2.5. Statistical Analysis

We ran descriptive and comparative analyses between the two groups on all the assessments, namely, perceived knowledge acquisition, life skills, and satisfaction with the intervention. We calculated means, standard deviations, and differences between the groups. Given the non-normal distribution, we performed nonparametric tests (Mann–Whitney). We also explored the difference in the grouping of life skills in five pairs. Bonferroni correction was used. The significance threshold was set at 0.05, and all the analyses were carried out using jamovi software version 1.8.

## 3. Results

### 3.1. Population Description

Participants were 225 students: 111 (29 males and 82 females; *M_age_* = 21.1 years, *SD* = 2.43) participated in the intervention featuring the *Selflife* tool; and 114 (30 males and 83 females; *M_age_* = 21.8 years, *SD* = 3.79) underwent a traditional intervention featuring the use of case studies. There was no significant difference in age or gender distribution between the two groups.

### 3.2. Effect on Knowledge Acquisition

No differences were observed between the two groups on knowledge acquisition (see Table 1). 

### 3.3. Effect on Life Skills

Participants in the *Selflife* group gave higher ratings than participants in the case studies group for five life skills: decision-making (*p* = 0.014), effective communication (*p* = 0.015), ability to relate to others (*p* = 0.031), creative thinking skills (*p* = 0.023), and stress management (*p* = 0.029) (see Table 2).

When we looked at the World Health Organization’s five pairs of life skills, the *Selflife* group had higher mean scores on all the pairs. These differences were significant for effective communication–interpersonal skills (*p* = 0.020), creative thinking–critical thinking (*p* = 0.041), self-awareness–empathy (*p* = 0.051), and emotion regulation–stress management (*p* = 0.038).

### 3.4. Effect on Satisfaction

For five of the six items probing satisfaction with the intervention, participants in the *Selflife* group gave higher ratings (see Table 3). More specifically, these students were significantly more satisfied with respect between participants (*p* = 0.004), the general atmosphere of the group (*p* = 0.003), mutual aid within the group (*p* = 0.014), and interactions between participants (*p* = 0.027), and had higher overall satisfaction (*p* = 0.014) than those in the case study group (*p* < 0.001) (see Table 1).

## 4. Discussion

In the present study, we carried out an initial assessment of a tool designed to prevent sexual violence: *Selflife.* We expected the use of this tool to have a more favorable impact than traditional, case study-based teaching on students’ learning and the development of their life skills in situations of sexual violence.

Our results showed that there was no statistically significant difference in the feeling of knowledge acquisition between students in the *Selflife* group and those in the case study group. We therefore concluded that our tool would be just as effective as case studies in transmitting knowledge about sexual violence.

As far as self-assessed life skills are concerned, *Selflife* students had a greater feeling than case study students that they had strengthened five of the ten life skills, namely, decision making, effective communication, interpersonal skills, creative thinking, and stress management. Finally, overall satisfaction was better in the *Selflife* group than in the case study group.

These initial results raise questions about the types of interventions that should be used to prevent sexual violence. Focusing on people’s resources and skills rather than their problems is the principle behind positive psychology. We therefore encourage the use of adaptive resources. This type of prevention program meets fundamental psychological needs, including the need for autonomy (i.e., need to choose, take initiatives, and freely decide on one’s behaviors or actions) [35]. We can even find a connection with the good lives model (GLM), which was developed for use with sexual offenders [36]. The GLM provides an intervention framework centered on the individual and that individual’s needs and resources, though without setting aside the principle of preventing reoffending. The GLM is based on the premise that it is important to develop knowledge, skills, and resources to meet our basic needs. Thus, developing a rich and meaningful life reduces the occurrence of deviant behaviors. This approach has also been shown to improve compliance with care, quality of life, and functional prognosis [36].

The GLM therefore provides a broader theoretical basis for primary prevention. *Selflife* is intended to prevent sexual violence by encouraging participants to work on situations that can lead to deviant behavior (perspective of future sex offender) and ways of avoiding or protecting themselves from this behavior (perspective of witness and/or victim). In other words, one of the strengths of this tool is that instead of targeting just one side of the violence, it allows participants to work on the different factors that promote sexual violence, which are not always explored in existing programs. Those programs that have proven to be effective [37], such as Safe Dates [38], Shifting Boundaries [39], and Violence Against Women Act [40], focus primarily on relationships between partners. Other programs that specifically target students tend to focus on the witnesses of sexual violence [41,42,43,44] and therefore do not specifically reduce the motivation of offenders to commit such acts.

These different elements suggest that this tool would limit perpetration, as well as reinforce strategies for coping with traumatic events. Reinforcing coping strategies (and therefore life skills) lessens the impact of traumatic events, making it possible to reduce the probability of PTSD [45].

Finally, it should also be noted that gender domestic violence is strongly linked to sexual violence [46], and that in France, domestic violence causes one victim every 3 days (125 deaths in 2020 [47]). Thus, we can hope that work on life skills development and knowledge on sexual violence would lead to reducing domestic violence.

The present pilot study had three limitations. First, although these are only preliminary results, the effect sizes were small (0.096–0.507), despite the reasonable sample size for a pilot study (225 participants). Second, we relied exclusively on a set of subjective and non-standardized measures. Standardized tools should be used in future research, in order to complement our self-report data and provide a more objective view of these results. These could include a knowledge assessment test or a recent life skills assessment scale [31]. Third, students in the *Selflife* group underwent a single session. Thus, it would be interesting to assess the impact of a more comprehensive program on possible changes in participants’ social representations. It should nevertheless be noted that, despite this limitation, we did observe changes in scores on the various scales we used.

Furthermore, our results did not allow us to ascertain whether the *Selflife* tool would indeed reduce acts of sexual violence and, thus, the occurrence of traumatic events. In the event of a wider deployment of this tool, it would be relevant to assess the occurrence of traumatic events within a larger sample and over a longer timeframe. In particular, the improvement of life skills by *Selflife* might allow traumatic events to be managed better, as these coping strategies limit the cognitive impact of such events [48].

## 5. Conclusions

This initial investigation of the *Selflife* tool for the prevention of sexual violence among healthcare students is, to our knowledge, the only study to have been carried out within a healthcare student population in France. *Selflife* improved life skills, increased knowledge about sexual violence just as much as the traditional teaching intervention did, and induced greater overall satisfaction with the intervention. These results should be replicated in studies using standardized measurement tools, with assessments beyond the end of the intervention, specifically to investigate the occurrence of traumatic events and the adoption of effective coping strategies. This study also raises the question of how such life skills development tools should be deployed in the future.

## Figures and Tables

**Table 1 ijerph-20-05198-t001:** Comparisons of pre- and postintervention scores (*Selflife* versus case study groups) on participants’ self-assessed knowledge acquisition.

	*Selflife* Group Mean (*SD*) *n* = 111	Case Studies Group Mean (*SD*) *n* = 114	Mann–Whitney *U*	*p* Value	Effect Size
* Self-assessed knowledge acquisition *					
“I felt comfortable talking about sexual health”	7.84 (1.97)	7.59 (2.49)	6255	0.881	0.011
“My participation did not allow me to acquire knowledge in the field of sexual violence”	3.41 (2.90)	3.30 (2.96)	5954	0.663	0.034
“Support gave me confidence to easily give my opinion in situations involving sexual violence”	8 (1.86)	7.51 (2.25)	5651	0.158	0.107
“It allowed me to reflect on possible situations that could be risky”	7.95 (2.26)	8.14 (2.27)	5938	0.411	0.062
“My participation did not change my representations/perceptions of sexual violence”	5.03 (3.04)	4.96 (3.07)	6140	0.786	0.021

**Table 2 ijerph-20-05198-t002:** Comparisons of pre- and postintervention scores (*Selflife* versus case study groups) on participants’ self-assessed life skills.

	*Selflife* Group Mean (*SD*) *n* = 111	Case Studies Group Mean (*SD*) *n* = 114	Mann–Whitney *U*	*p* Value	Effect Size
* Self-assessed life skills *					
Solving problematic situations	6.85 (2.22)	7.08 (2.06)	5947	0.498	0.052
Knowing how to make decisions	7.22 (2.36)	6.58 (2.32)	5052	0.014	0.187
Communicating effectively	7.85 (1.94)	7.08 (2.43)	5064	0.015	0.185
Being skillful in my relationships with others	7.66 (1.93)	7.06 (2.21)	5244	0.031	0.164
Adopting creative thinking	6.59 (2.83)	5.89 (2.64)	5031	0.023	0.176
Developing critical thinking skills	8.40 (1.45)	8.15 (1.78)	5799	0.317	0.075
Developing self-awareness	7.72 (2.36)	7.27 (2.41)	5533	0.122	0.118
Having empathy for others	8.58 (1.93)	8.20 (1.96)	5479	0.090	0.126
Knowing how to manage stress	6.68 (2.50)	5.76 (3.0)	5221	0.029	0.168
Knowing how to manage my emotions	6.78 (2.46)	5.97 (2.92)	5332	0.051	0.150

**Table 3 ijerph-20-05198-t003:** Comparisons of pre- and postintervention scores (*Selflife* versus case study groups) on participants’ satisfaction with participation.

	*Selflife* Group Mean (*SD*) *n* = 111	Case Studies Group Mean (*SD*) *n* = 114	Mann–Whitney *U*	*p* Value	Effect Size
* Satisfaction with participation *					
Respect between participants	1.03 (0.21)	1.12 (0.36)	5669	0.004	0.096
General atmosphere of the group	1.26 (0.48)	1.51 (0.70)	5100	0.003	0.187
Self-help within the group	1.45 (0.63)	1.69 (0.75)	5106	0.014	0.171
The clarity of the information given	1.20 (0.46)	1.19 (0.43)	6105	1	0
The interactions between participants	1.52 (0.70)	1.77 (0.84)	5152	0.027	0.156
Overall satisfaction with participation	1.45 (0.67)	1.80 (0.80)	4487	<0.001	0.507

## Data Availability

Data from this study can be made available upon request to the corresponding author.

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
