# Peer review of "Selflife: A Life Skills Development Tool to Prevent Sexual Violence among Healthcare Students"

_ijerph, 2023, doi:10.3390/ijerph20065198_

Round 1
Reviewer 1 Report
Line 36 - It is necessary to define the concept of "Sexual Violence" used in this study. And it is important to check that the surveys [6] referred to use the same definition.
Line 42 - Is there any kind of response from the authorities and can you cite any recent scientific research?
Lines 51 and 81 - Are the reports of observatories and associations themselves scientific studies?
Line 96 - I could cite some tools here.
Line 135 - It would be interesting to provide a description of case studies similar to section 2.3.
Line 147 - How has the Questionnaire been validated?
Author Response
Dear reviewer, thank you for your time in reading and evaluating our manuscript. We found the comments helpful and believe that the revisions have improved it.
Here are the improvements made regarding the points you noted:
Reviewer #1: Line 36 - It is necessary to define the concept of "Sexual Violence" used in this study.
Answer : We have added a sentence regarding the definition of sexual violence for the cited study.
“In this study, sexual violence is defined as any act related to rape, attempted rape, sexual assault, or coercion related to sexual acts (e.g., being forced to watch sex films). These events could occurred in the context of school, work, public spaces (broadly defined), in the context of a couple or ex-spouse relationship (including boyfriends or non-cohabiting relationships), and in the context of family and close relationships. Excluding harassment and exhibitionism.”
And it is important to check that the surveys [6] referred to use the same definition.
Answer : We have clarified the definition of sexual violence of the survey by adding a new sentence :
“For example, in the study of Williams et al. [6], where they explored unwanted sexual activities with pressured (by friendship or romantic relationship) or another person threatened to use or used physical force, or when they did not want to because they were drunk or on drugs, they observed that 12,45% of students have experiment in the last 12 months a sexual violence.”
Line 42 - Is there any kind of response from the authorities and can you cite any recent scientific research?
Answer : We have added this sentence :
“Given its prevalence, sexual violence is therefore a major public health problem [4,5]Which persists despite enhancement of research in this field (three more times of articles on 2022 than 2010 about “sexual violence” on Google scholar) or political responses (such as the European program DAPHNE which aims to reduce gender violence and has a budget of 80 million euros for the period 2021-2027 [6]).”
Lines 51 and 81 - Are the reports of observatories and associations themselves scientific studies?
Answer : We agree with the reviewer that these documents may not be scientific. However, in the absence of other information available in France, we can only have these types of sources of information. However, the more scientific studies are cited in other parts of our manuscript (ex : Williams et al.) and are consistent with the observations observatories and associations. These last reports are essentially used for a French description considering the population of this study. These reports give us an indication from an epidemiological point of view, which are certainly elements of the field, but which testify to sexual violence in the student environment.
Line 96 - I could cite some tools here.
Answer : Indeed, our sentence was not precise enough. We have chosen to remove it
Line 135 - It would be interesting to provide a description of case studies similar to section 2.3.
Answer : We have added a paragraph 2.4 to better explain what was done in the case study group.
“For the "case studies" teaching method, we chose to present examples of sexual violence in the student population based on facts reported in the national press. The objective was also to encourage exchanges within the group of students about the case presented, while mobilizing their psychosocial skills, by developing their critical thinking skills, for example, or by probing them about possible resources in the face of sexual violence.“
Line 147 - How has the Questionnaire been validated?
Answer : As we mentioned in our limitations, our questionnaires are not standardized. We have therefore emphasized this point in the limitations by changing a sentence.
“We relied exclusively on a set of subjective and non-standardized measures. Standardized tools should be used in future research, in order to complement our self-report data and provide a more objective view of these results.

Reviewer 2 Report
Dear authors,
The topic you present is interesting and relevant in today's context.
Nevertheless, there are some points in the article that can be discussed or corrected.
The introductory part could be supplemented with information about perpetrators of sexual aggression (you mentioned them in your conclusion).
Check the number of study participants, as it is different (see lines 131 and 186)
Table 1 in the article is very large. I suggest you think about how to shrink it.
The graphs in Figure 1 are very uninformative. I suggest to correct them.
In the discussion section (see lines 223-226), it would be great if the authors could substantiate this phrase with scientific insights.
I would also debate the necessity of the first sentence in the conclusion part.
Author Response
Dear reviewer, thank you for your time in reading and evaluating our manuscript. We found the comments helpful and believe that the revisions have improved it.
Here are the improvements made regarding the points you noted:
Reviewer #2: Dear authors,
The topic you present is interesting and relevant in today's context.
Nevertheless, there are some points in the article that can be discussed or corrected.
The introductory part could be supplemented with information about perpetrators of sexual aggression (you mentioned them in your conclusion).
Answer : We thank the reviewer for this remark and have added two paragraphs describing the perpetrators of sexual aggression.
“In nine cases out of ten, the victims know their aggressor, who can be a boyfriend, romantic partner or another student [11]. In the majority of cases, the facts would have taken place in an festive context or within the student residence [12]. Adherence to rape myths encourages sexual violence, while minimizing the responsibility and guilt of the perpetrators [13–16]. Some men may interpret a woman's friendly interest as sexual interest, her lack of consent as feigned resistance, her stunned response to an assault as a form of tacit acceptance. Studies also show a strong link between being sexually abused as a child or teenager and becoming a perpetrator in turn [17,18]. There is probably no typical portrait of a perpetrator of sexual violence. Young adult students are at a particularly high risk for violence. The combination of all three types of factors predicts the severity of aggression.”
“Moreover sexual violence prevention efforts must focus on both the victims and the potential perpetrators [18]. To do this, work must be done on learning about the conditions under which sexual violence occurs, social representations, and finally on developing the skills to deal with an experience of sexual violence.”
Check the number of study participants, as it is different (see lines 131 and 186)
Answer : Indeed, this was a typo error, as we do have 225 students who were included in this study. This has been changed and harmonized in the document.
Table 1 in the article is very large. I suggest you think about how to shrink it.
Answer : Following this recommendation, we have now divided our table into three new table:
- Table 1. Comparisons of pre- and postintervention scores (Selflife versus case study groups) on participants' self-assessed knowledge acquisition
- Table 2. Comparisons of pre- and postintervention scores (Selflife versus case study groups) on participants' self-assessed life skills
- Table 3. Comparisons of pre- and postintervention scores (Selflife versus case study groups) on participants' satisfaction with participation.
The graphs in Figure 1 are very uninformative. I suggest to correct them.
Answer : Thank you for this remark. We have decided to delete the figure 1 considering the fact that the elements referring to it are already in the “Result part”.
In the discussion section (see lines 223-226), it would be great if the authors could substantiate this phrase with scientific insights.
Answer : We chose to add a sentence in the "objectives" section before pointing out that the classic case study mode of learning has already proven itself in medicine with a new reference :
“We hypothesized that a specific prevention tool such as Selflife can have a more favorable impact on these dimensions than an intervention that solely involves the presentation of case studies which is already an effective teaching modality in medical education.”
I would also debate the necessity of the first sentence in the conclusion part.
Answer : We decided to reformulate this last sentence : “This initial investigation of the Selflife tool for the prevention of sexual violence among healthcare students is, to our knowledge, the only study to have been carried out within a healthcare student population in France”.

Reviewer 3 Report
Gender violence is not only a "mayor public health issue", we need to foster the influence -negative- of the ideas of chistian religion, political parties (rigth wind as in some of the european countries including France an social ideas about higher position of men on women and the idea of patriachal organization of the society in a neoliberal capitalist society, so you need to explain better the meaning of "gender violence as an expression of power" and why victims are mostly women and perpetrators are men.
The impact of violence on women in France is more dangerous acording to EU Reports that you should include.
You should not mix objetives and hypotheses. My view is that according to your method you must employ onlu objetives and ensure the link between them and the results.
Gender violence is a social, educational and political issue mainly and you have to read some reviews of the EU about Gender Violence Prevention Educational Programmes (Daphne).
Author Response
Dear reviewer, thank you for your time in reading and evaluating our manuscript. We found the comments helpful and believe that the revisions have improved it.
Here are the improvements made regarding the points you noted
Reviewer #3: Gender violence is not only a "mayor public health issue", we need to foster the influence -negative- of the ideas of chistian religion, political parties (rigth wind as in some of the european countries including France an social ideas about higher position of men on women and the idea of patriachal organization of the society in a neoliberal capitalist society, so you need to explain better the meaning of "gender violence as an expression of power" and why victims are mostly women and perpetrators are men. The impact of violence on women in France is more dangerous acording to EU Reports that you should include.
Answer :
We are indeed sensitive to the socio-cultural environments that can support gender-based and sexual violence. Nevertheless, the aim of the project is to raise awareness of social representations and attitudes towards sexual violence (including gender-based violence) in a broader perspective encompassing psychosocial skills and therefore not only focused on sexual violence against women.
We have added more specific information on victims and characterizes of perpetrators of sexual violence based on national studies on our introduction and we added a new sentence on gender domestic violence on our discussion section as a possible perspective of this study.
“Finally, it should also be noted that gender domestic violence is strongly linked to sexual violence [37], and that in France domestic violence causes one victim every 3 days (125 deaths in 2020 [38]). Thus, we can hope that work on life skills development and knowledge on sexual violence would allow to reduce domestic violence.”
You should not mix objetives and hypotheses. My view is that according to your method you must employ onlu objetives and ensure the link between them and the results.
Answer : We have reworked the hypotheses and objectives section on the introduction by removing the hypotheses to focus on the objectives
“We hypothesized that the Selflife intervention would be associated with (a) a feeling of acquiring knowledge about the social representations of sexual violence at least equivalent to traditional teaching; (b) a better development of their life skills and; (c) a better satisfaction with their participation in this teaching. For this we will use a comparison postintervention between an intervention using Selflife and an intervention using the presentation of case studies which is already an effective teaching modality in medical education [26].”
Gender violence is a social, educational and political issue mainly and you have to read some reviews of the EU about Gender Violence Prevention Educational Programmes (Daphne).
Answer : thank you for this suggestion, we have added in the introduction some elements concerning the Daphne program and we added a sentence in the discussion section in view of perspectives.
Introduction section : “Given its prevalence, sexual violence is therefore a major public health problem [4,5] which persists despite enhancement of research in this field (three more times of articles on 2022 than 2010 about “sexual violence” on Google scholar) or political responses (such as the European program DAPHNE which aims to reduce gender violence and has a budget of 80 million euros for the period 2021-2027 [6,7]).”
Discussion section: “Finally, it should also be noted that gender domestic violence is strongly linked to sexual violence [37], and that in France domestic violence causes one victim every 3 days (125 deaths in 2020 [38]). Thus, we can hope that work on life skills development and knowledge on sexual violence would allow to reduce domestic violence.”

Round 2
Reviewer 2 Report
Dear authors, I think you have taken into account the advice and made quality corrections to the article